# Structural Behavior Evaluation of Reinforced Concrete Using the Fiber-Reinforced Polymer Strengthening Method

**DOI:** 10.3390/polym13050780

**Published:** 2021-03-04

**Authors:** Tae-Kyun Kim, Jong-Sup Park, Sang-Hyun Kim, Woo-Tai Jung

**Affiliations:** Structural Engineering Research Institute, Korea Institute of Civil Engineering and Building Technology, Goyang 10223, Korea; jSpark1@kict.re.kr (J.-S.P.); kimsanghyun@kict.re.kr (S.-H.K.); woody@kict.re.kr (W.-T.J.)

**Keywords:** fiber reinforced polymer, near surface mounted, external bonding, section enlargement, external prestressing, structural polymer, nanofiller

## Abstract

Reinforced concrete (RC) structures age with time, which results in performance degradation and cracks. These performance degradations do not recover easily, but a performance higher than the existing structures can be expected through reinforcement. There are various reinforcement methods for RC structures. This study selected four reinforcement methods: near-surface mounting (NSM), external prestressing (EP), external bonding (EB), and section enlargement (SE). In the past, steel bars were often used as reinforcements. However, this study uses fiber-reinforced polymer (FRP), which is an alternative to steel bars owing to its high tensile strength, and its non-corrosive and lightweight properties. It is a basic strengthening material, along with a carbon-fiber-reinforced polymer (CFRP) and glass-fiber-reinforced polymer (GFRP) in bar and sheet forms. Various strengthening materials such as a CFRP, GFRP, and prestressing (PS) strand are applied to the NSM, EP, EB, and SE methods, followed by flexural experiments. In addition, changes in the ductility of the RC structures were examined. The concrete EP and near-surface mounting prestressing (NSM(P)) methods have a stiffness that is almost double the non-strengthened specimen. However, because the EP and EB methods are brittle, the NSM(P) method with ductile behavior is considered the most effective.

## 1. Introduction

Reinforced concrete (RC) structures are being applied worldwide to a variety of infrastructures owing to their excellent structural performance, durability, fire resistance, and economic efficiency. However, concrete is a brittle material and its disadvantages include cracking, partial breakage, and its heavy weight [1]. The cracking, stress, and strain of RC appears to be different depending on the member type, applied load, member shape, dimensions, the arrangement of the steel bars, and the properties of the concrete and steel bars [1]. Numerous structures were designed and constructed during the 1970s. As a result, existing structures that are more than 50 years old have safety and usability problems with the rising importance of maintenance and growing reinforcement costs [2]. Once RC structures show a rapid performance degradation, such as cracks, concrete strength reduction, and steel bar corrosion due to the deterioration and aging of the material, it is difficult for them to be recovered by themselves [3,4]. The performance degradation can also cause economic and human damage directly and indirectly [5]. To solve these problems and improve the reduced performance of the concrete structures, accurate reinforcement and precise design methods are required. However, except for the USA, Japan, and Europe, reinforcement design or guidelines are not clearly defined in most countries, including South Korea [6,7,8]. Therefore, to provide clear designs and guidelines in the future, a database of diverse studies is required [6,7]. To improve the performance of the structures, various reinforcement methods are being applied. Strengthening methods for RC structures include: near-surface mounted (NSM), which buries the reinforcements in concrete grooves and fillers [9,10,11,12]; external prestressing (EP), which uses a prestress force through anchorage and a strand outside the structure [13,14,15]; external bonding (EB), which reinforces by bonding by using fiber sheets and plate elements on the tensile side of concrete [16,17]; and section enlargement (SE), which also uses reinforcements in the existing structures and enlarges the cross section [18,19]. Among the existing studies that are related to reinforcement, Lee et al. [20] evaluated the flexural performance of structures with the single NSM method according to the number of steel bars. Barros et al. [21] compared the performances of the EBR and EBROG methods according to the number of reinforcement layers and the number of buried grooves. Fan et al. [22] evaluated the explosion-load resistance based on the EP. Chen et al. [23] evaluated the performance of the structures according to the reinforcement direction and the length of the CFRP sheets regarding the EB. Li et al. [24] evaluated the sectional extension reinforcement and performance according to the reinforcement amount of the D14 steel bar. As shown in these examples, domestic and overseas studies regarding single methods and the characteristics of the reinforcement amount such as the reinforcement ratio and the number of reinforcement layers are being actively conducted [20,21,22,23,24]. To distinguish from existing studies, this study compares and analyzes the structural performances for several strengthening methods by using multiple strengthening materials. In addition, to realize material strengthening, fiber-reinforced polymers (FRPs) can replace steels, the use of which is reported in existing literature. Moreover, FRPs are axial particulates embedded in fitting matrices. The advantages of FRCs over unreinforced materials are well known and their characteristics are useful in many areas. FRC applications are being increasingly researched in the fields of aerospace, medicine, and construction science [25,26,27]. FRP composite, including fibrous porous media, has been widely used in many fields of life. Besides the NSM, EP, EB, and SE methods, the fractal method is a very important approach, which can be used to investigate the physical properties of the FRP composite, including fibrous porous media [28,29]. FRPs have gained global popularity owing to their advantages, such as high tensile strength, non-corrosive behavior, and low weight. Popular types of FRPs include the CFRP and glass-fiber-reinforced polymer (GFRP), which uses glass fiber as reinforcement [30]. In fact, many investments are being made in FRPs not only in the material industry, but also in the development of applied technologies due to their high usability and potential marketability in the construction industries in developed countries, which includes the US, Europe, and Japan [31]. However, the PS tendons and FRPs used in this study can cause brittle behavior of the structures, as shown in Figure 1 [32]. It can be observed that RC structures exhibit ductile behavior since they have a yield point on the reinforcing bar that is used as reinforcement, but the weakening of the PSC structure or the uncertain yield point of the FRP material causes brittle fractures [33,34]. To verify the ductility of the existing RC structures and the RC structures reinforced with brittle materials, the ductility index of structures was represented by using the ductility evaluation method that was proposed by Jeong in 1994. Based on this result, the energy ratio that was studied by Grace in 1998 was derived and analyzed for ductile, semi-ductile, and brittle properties [35].

Therefore, this study analyzes the structural performance by conducting flexural experiments with various strengthening materials such as CFRP, GFRP, and PS strands while applying strengthening methods such as NSM, EP, EB, and SE. In addition, the changes in the ductility after strengthening the RC structures were examined. The results of this study can be used as basic data for achieving the safety of reinforcement design in the future. Lastly, the most critical advantages of the NSM, EP, EB, and SE methods determined from this study are as follows: the NSM method protects the reinforcement material with the burial effect, the EP method introduces external force, the EB method enables easy construction, and the SE method increases the bending moment due to the rise of the neutral axis. These methods are economical and safe, and the structural performance is expected to improve through their application.

## 2. Background of Fiber-Reinforced Polymer (FRP)

In the history of composite materials, FRPs were developed over six decades ago for applications in leisure activities. With the development of material technology and the passage of time, fibers such as aramid, carbon, and glass have been developed for high functionality, mechanical properties, high durability, and weight reduction [36]. During the 1960s, composite materials were mainly used in national defense and aerospace due to the high costs of composite materials. In the 1970s, since the mass production of composite materials was possible, the cost was lowered, and sporting goods that could be easily used by the public were expanded. However, from the late 1980s to early 1990s, the defense market suffered a recession, though the prices of composite materials were further reduced by the virtue of continued growth. Consequently, developed countries around the world expanded the applications of composite materials to infrastructures from the mid- to late-1990s [37]. During this period, corporations and governments supported various studies and projects for composite materials that are related to construction. In the 2000s, composite materials were commercialized or applied as next-generation construction materials in the construction industry to improve the performance of concrete infrastructure facilities [26].

Composite materials refer to the combination of two or more types of materials. Conceptually, composite materials are artificially manufactured materials with a higher performance than the existing materials while physically and chemically maintaining the original phase of each material, even after being combined with two or more types of materials [37]. Composite materials can be largely classified into particle-reinforced materials, fiber-reinforced materials, and structure-reinforced materials. Composite materials that are used in construction generally represent the combinations of fibers with excellent mechanical properties and polymers that constrain, shape, and transmit stress to the fibers. For construction fibers, glass fibers and carbon fibers are widely used. Thus, based on their fiber type, FRPs can be classified as CFRP, GFRP, and aramid-fiber-reinforced polymer (AFRP) [26].

Although many studies have been conducted on FRPs, FRPs are applied to construction designs only in the US, Europe, and Japan, and there are a number of countries where FRPs are not yet standardized [26].

### 2.1. Carbon-Fiber-Reinforced Polymer (CFRP)

CFRP is a material that exhibits high elasticity and strength by adding carbon fibers to plastics, and carbon fiber composite materials are made of carbon fiber as a reinforcing material and a matrix resin that is combined with plastics [32]. In comparison to iron, CFRP has a strength that is 10 times higher and an elastic modulus that is seven times higher, but its weight is only 25% that of iron [32]. CFRPs are non-corrosive materials with excellent wear resistance, heat resistance, durability, and impact resistance, and they have been used as a core component in composite materials in the semiconductor, aerospace, and aerospace industries. Furthermore, the use of CFRPs has been increased in the automotive industry owing to the weight reduction of motor vehicles, which is also the reason CFRPs are applied in sports. Recently, the applications of CFRPs in construction materials and medical industries have continued to expand.

### 2.2. Glass-Fiber-Reinforced Polymer (GFRP)

GFRP is a plastic that is reinforced with glass fiber and it is processed by an unsaturated polyester with a diameter of 0.1 mm or less. GFRP is lighter than aluminum, and it is stronger, lighter, and harder than general iron; thus, it can resist an external impact and it has a very high tensile strength [30]. Furthermore, GFRPs have advantages such as the convenience of processing and resistance to rusting, but their disadvantage is that they cannot be used at high temperatures. GFRPs are applied to construction materials, boats, ski products, helmets, and parts for automobiles and aircraft.

## 3. Reinforcement Techniques

### 3.1. Near-Surface Mounted

NSM is made by digging grooves and placing reinforcements in the existing structures. In the past, reinforcement was mainly carried out with steel bars, but reinforcements that use FRP materials have been recently applied [9]. If tension is required, anchorages can also be installed depending on the situation. When the construction is completed, it is filled with an epoxy or grout injection. NSMs must be allowed to behave integrally with the existing structure in the reinforced position. The greatest advantage of NSM is that the reinforcements are not exposed to the outside, and it is safe from the various environmental deterioration phenomena over time due to the burial effect.

### 3.2. External Prestressing

EP is applicable to a variety of structures that are made of concrete or steel such as columns, bottom plates, and beams. EP is a method that introduces a prestress force by installing anchorages on existing structures, and it places new PS tension members to improve the stress state and load carrying capacity of the existing structures [14]. However, this method can cause rust due to moisture and carbon dioxide with time since the steel wires are exposed to the outside, which can be replaced with FRPs.

### 3.3. External Bonding

Fiber EB can secure the safety of structures and maintain them in a healthy condition. It has the advantage of allowing smooth transportation in the process of bridge reinforcement [23]. Furthermore, the FRP bonding method can enhance the structural performance by integrating fibers with the existing members in the form of a sheet and plate. CFRP, GFRP, and AFRP are used as the materials for EB and they are mainly applicable to the girder and slab structures of concrete bridges. Furthermore, EB must enable the integral behaviors of fibers and structures.

### 3.4. Section Enlargement

SE has mainly been used to reinforce the seismic performance of RC column structures, and the studies on its applications to general structures are insufficient. SE is applied to concrete bridges for increasing the bending moment that accompanies the rise of the neutral axis by improving the load-bearing performance against compression and shearing of the top plate [19]. Furthermore, it can be applied to members such as columns, foundations, beams, and slabs as a strengthening method by expanding the sections of the existing members. It is a method of placing and pouring fiber-reinforced concrete and steel bars. In addition, integration of the interface between the base member and extension member is crucial.

## 4. Test Conditions and Methods

### 4.1. Concrete Mixture Properties

Table 1 presents that the target strength was set by using the concrete mix design and the test specimen was prepared specifically for this test [38]. The compressive strengths of concrete after 28 days are 20 and 40 MPa. The physical properties are identical in the 40 MPa high-strength mix, except for the blast-furnace slag cement. For the cement, a class 1 product was used, and the same materials were used for water, coarse and fine aggregates, and fly ash.

### 4.2. Specimen Details

Figure 2 shows the drawing and cross-sectional area of the RC beam that was used in this study, which has a total length of 6400 mm and a height of 600 mm. For the top and bottom compression steel bars, three pieces of D22 (steel bar diameter) and D19 were used, respectively. For the band reinforcement, D10 was used. Figure 3 and Table 2 show the strengthening method, the number of specimens, and the nomenclature of the specimen name. There are 11 specimens in total, which are composed of two control specimens and nine reinforced specimens, and they have the same RC structure. The compressive strengths of the concrete were divided into 20 and 40 MPa, and four strengthening methods were applied: NSM, EP, EB, and SE. The strengthening materials were classified into the CFRP bar, PS Strand, CFRP sheet, and GFRP bar, and there was one CFRP bar, two PS strands, two layers of CFRP sheets, and six GFRP bars, respectively. Table 3 outlines the material properties, i.e., Young’s modulus, yield stress, and ultimate stress, of these strengthening materials (TDS-530 data logger, Tokyo Sokki Kenkyujo, Tokyo, Japan). Furthermore, Figure 4 illustrates the specimen strengthening methods. The NSM method installs anchorages through grooving, and it inserts CFRP steel bars and fills the epoxy. The EP method installs anchorages to the bottom outside of the specimen and uses two strands of SWPC7B (strands and wires for prestressed concrete of 7B type) with a 12.7 mm diameter. EB is strengthened with two layers of CFRP with a width of 100 mm, a thickness of 2 mm, and a length of 6 m. Finally, the SE method places six GFRP steel bars at fixed intervals and strengthens the entire bottom surface that measures 6400 × 400 mm^2^.

### 4.3. Flexural Strength Test Setup

Figure 5 shows the flexural experiment method. A TDS-530 data logger (Tokyo Sokki Kenkyujo, Tokyo, Japan) was used for the measurements. In addition, concrete and reinforcement strain gauges were used to measure the strain. The concrete gauges were attached to the top, middle, and bottom surfaces of the central part, and at the 3/4 point from the bottom. Reinforcement gauges were installed at the central upper compression steel bar, 1500 mm to the left and right from the center; they lowered the tension reinforcement. In addition, to measure the displacement according to the load, linear variable differential transformers (LVDTs) were installed at the same positions as the tension steel bar gauges, which were 200 mm away from both ends, and the load was applied at 500 mm to the left and right from the central top surface. In addition, the load was controlled to 30 mm at the rate of 0.03 mm/s, and the displacement control of 0.1 mm/s was performed later. The universal testing machine (UTM) used in this study was a 200 ton-grade device (Galdabini, Varese, Italy). Thus, based on the flexural experiment, the load–displacement graph is determined using the SigmaPlot software program (SYSTAT, San Jose, CA, USA).

## 5. Test Results

Table 4 and Table 5 show the data results for each load of the specimens, and Figure 6, Figure 7 and Figure 8 show the results of displacement using the LVDT of the specimens. Specifically, Figure 6 shows the strength, Figure 7 shows the concrete strength, and Figure 8 shows the results based on all specimens. Figure 9, Figure 10, Figure 11 and Figure 12 show the derived strain results, which compare the strain according to the material of the specimen. In particular, Figure 9 and Figure 10 present the results for the concrete and Figure 11 and Figure 12 show the rebar strain.

### 5.1. Comparison Between Crack, Yield, and Ultimate Loads

Table 4 compares the design values that are presented by the manufacturer and the actual experimental values for the crack load ($P_{cr}$), yield load ($P_{y}$), and ultimate load ($P_{u}$) for each specimen. In the case of the NSM crack load, every experimental value is lower by approximately 8–30% compared to the design value. In contrast to the crack load, the experimental value of the yield load is higher by approximately 10–30%. For the ultimate load, the design and experimental values are similar.

In EP, the experimental values for the initial crack load and yield load increase by approximately 10%, and the experimental values for the ultimate load increase sharply by approximately 25–40%. In EB, the experimental values for the crack load and yield load are higher by approximately 40%. However, the experimental values for the ultimate load sharply decrease by more than 50% compared to the design value. Finally, in the case of SE, the experimental values for the crack load, yield load, and ultimate load were higher by approximately 20% compared to the design values. Thus, NSM shows similar design and experimental values, whereas the EP, EB, and SE methods show significant differences between the design and experimental values. This is because the strengthening design is not clearly defined in South Korea, and the EB method shows bonding failure in the actual experiment. Furthermore, in the case of the SE method, it is believed that the structure did not exhibit its proper performance owing to the inability to perform the integral behavior of the extension part and the existing structure.

### 5.2. Load–Displacement Results for Different Strengthening Methods

Table 5 shows the load–displacement results. Figure 6 illustrates the methods, Figure 7 displays the methods according to the concrete strength, and Figure 8 presents the results of all the specimens.

Figure 6a shows the load–displacement curve of the non-strengthened RC specimens. The maximum loads of R4C and R2C are 163.5 and 155.4 kN, respectively, and the maximum deflections are 82.68 and 159.87 mm, respectively. Furthermore, the stiffness of the yield load of the R4C is higher by approximately 15%. Thus, it can be observed that the concrete strength affects the stiffness because the material properties other than the concrete strength are the same in the specimen specifications. Figure 6b shows the NSM(P) load–displacement curve where the maximum loads of R4NSP and R2NSP are 337.7 and 229.5 kN, respectively, and the maximum deflections are 155.98 and 110.16 mm, respectively. Thus, the maximum loads equaled 106% and 47% higher compared to those corresponding to the non-strengthened specimen. In addition, the stiffness values increased by 100% and 50%, respectively. Figure 6c shows the NSM(N) load–displacement curves that facilitate strength comparison between products sourced from the H and S manufacturers. The maximum loads are 180.1 and 206.9 kN, respectively, whereas the maximum deflections are 50.61 and 99.84 mm, respectively. Furthermore, the H strengthening material shows a similar trend to the non-strengthened specimen. However, the strengthening materials of the S manufacturer are similar before the yield, but after yielding, it shows more ductile behavior than the strengthening material of the H manufacturer, but the maximum deflection is smaller. Figure 6d shows the EP load–displacement curve. The maximum loads of R4EPP and R2EPP are 291.4 and 252.5 kN, respectively, and the maximum deflections are 39.93 and 46.05 mm, respectively. Compared to the non-strengthened specimen, the maximum loads of the above-mentioned specimens demonstrated increases of 78% and 62%, respectively. The stiffness increased twice for each specimen. However, the structure due to the prestress force shows more brittle behavior than the RC. Figure 6e shows the EB load–displacement curve. The maximum loads of R4EBN and R2EBN are 252.5 and 232 kN, respectively. The maximum deflections are 46.05 and 44.49 mm, respectively. Compared to the non-strengthened specimen, the load increases by 42% and 85%, respectively, whereas the stiffness increases by 50% and 55%, respectively. Figure 6f shows the SE load–displacement curve, the maximum load is 243.9 kN, and the maximum deflection is 96.63 mm. Compared to the non-strengthened specimen, the maximum load increased by 57% and the stiffness until the yield load increased by approximately 60%. In the existing literature, Lee et al. [21] presented that the stiffness to yield is increased by about 40% in the NSM reinforced test sample compared to the control test sample, Wang et al. [14] presented that the stiffness of the test specimens increases by approximately 50 to 70% until reaching the yield load using the EP method according to the prestressing force compared to the control specimen, Mostofinejad et al. [16] stated that the EB method increases the stiffness by approximately 10 to 40% compared to the control specimen, and Kim et al. [20] confirmed that the SE method can yield almost twice the stiffness of the control test specimen through a combination of mortar, steel plate, and anchor. As such, the results of the present study and previous literature demonstrate that the reinforcement methods tend to yield a higher stiffness than the control test specimen, and similarly, the stiffness tends to increase as prestressing is added.

Figure 7 compares the strengthening methods under the different concrete strengths. Figure 7a shows the load–displacement curves for each method with a concrete strength of 40 MPa. The specimen that exhibits the maximum load is R4NSP with 337.7 kN, which also shows the highest maximum load. Furthermore, the EP shows the largest stiffness until the yield load is reached, but the difference with R4NSP is not significant. Hence, with a concrete strength of 40 MPa, NSM appears to be the best method. Figure 7b shows the load–displacement curves of each method with a concrete strength of 20 MPa. R2EBN shows a maximum load of 287.7 kN, and R2NSP shows a maximum deflection of 110.16 mm. The stiffness values are similar among the EP, EB, and SE methods, but EP and EB display brittle behavior after yielding. Therefore, by having a concrete strength of 20 MPa, the SE and NSM(P) methods are the most efficient. Figure 8 shows the load–displacement curves for all the specimens. The methods that show the highest maximum load, maximum deflection, and stiffness are R4NSP, R4NSP, and R4EPP, respectively. Hence, when all the conditions are considered, the NSM(P) of the R4NSP specimen is considered the most efficient method in this study. The concrete EP and NSM(P) methods provide almost double the stiffness of the non-strengthened specimen. However, because the EP and EB methods result in brittle behavior, the NSM(P) method with ductile behavior is considered the most effective. For this reason, the characteristics of the materials reinforced in the structure appear to be the greatest, as shown in Figure 1. CFRP and prestressing steel have approximately 3.5- and 2.5-times higher stress than reinforced steel, respectively. For the stress, when external loads such as compression, tension, bending, and torsion are applied to the material, the resistance force generated inside the material is called stress. The stress increases with the external force, but there exists a limit. Additionally, since the maximum strain of reinforced steel is approximately four and two times higher than that of CFRP and prestressing steel, respectively, it is critical that it is deformed while maintaining the inherent properties of the material under various loads. Hence, CFRP and prestressing steel materials exhibit brittle behavior as they have high stress and low strain, and reinforced steel materials can exhibit ductile behavior as they have relatively low stress but high strain.

### 5.3. Steel Strain

Figure 9 shows the load–strain curves for the steel compression and the tension of each strengthening method according to the compressive strength of the concrete. Figure 10 shows the load–strain curves for all the specimens. The steel bar that was used in this study had a yield strength of 400 MPa. Thus, it is expected to yield when the strain of the steel bar is approximately 0.002 or higher. The non-strengthened specimen yielded under a load of approximately 140 kN. The specimen that showed the highest yield point was R4EEP and the specimen that showed the lowest yield point was R2NSN(S). The loads of these two specimens are 270 and 154.2 kN, respectively, which improved by 90% and 10%, respectively, compared to the non-strengthened specimen.

### 5.4. Concrete Strain

Figure 11 shows the load–strain curves for concrete compression and tension for each strengthening method according to the compressive strength of concrete. Figure 12 shows the load–strain curves for all the specimens. The compressive failure strain of concrete was 0.003. When a compressive failure of concrete occurs, a brittle failure phenomenon occurs, in which the structure collapses rapidly. Therefore, RC structures should pursue ductile failure behavior for structural safety. Figure 12 shows that every specimen reaches the maximum load before reaching the compressive failure strain of 0.003.

### 5.5. Suggestion of Design Plans for Each Reinforcement Technique

This study intends to suggest considerations and improvements to problems when designing each reinforcement technique through a research experiment. In the NSM method, when applying the near-surface mounted strengthening method to a structure, it is judged that it is necessary to proceed with the formation of groove and anchor, and in the case of inserting a fixing device into the anchorage and installing an anchor, a regulation on the depth of anchor burial is required. It is also believed that additional consideration for prestressing loss is needed. The EP method requires an experimental study to prevent fracture of the anchorage and establish a detailed design plan for the anchorage, after which the spacing and effective depth of the anchor can be considered. The EB method is designed in consideration of the debonding of the structure and the FRP reinforcement, and an experiment is required to test the interfacial failure energy of the interfacial adhesion between the reinforcement and the structure. Additionally, the permissible range of anchor installation and the specification of the installation standard is required. Lastly, for the SE method, it is necessary to quantitatively evaluate the degree of adhesion to the reference structure through the adhesion strength test of the new mortar in the design process, as well as to consider the anchor installation or to add admixtures and additional measures to improve the mortar adhesion performance. Additionally, when reinforcing FRP, it is necessary to evaluate the behavior with the structure. In conclusion, sustainable and safe reinforcement methods can be derived by considering various designs for each construction method.

## 6. Ductility Evaluation of Reinforced Concrete

### 6.1. Ductility Theory

Ductility is a qualitative concept that refers to the degree of deformation until failure in terms of the structural members, structural sections, and materials [29]. It is the most important item that should be considered together with the stiffness and ultimate strength when evaluating the basic safety of the structures. The measures for evaluating the ductility include the ductility index and ductility ratio, which are expressed as follows [29]:(1)$$\mu_{\Delta }=\frac{\Delta_{u}}{\Delta_{y}},\;\mu_{\theta }=\frac{\theta_{u}}{\theta_{y}},\;\mu_{\emptyset }=\frac{\emptyset_{u}}{\emptyset_{y}}$$

The quantitative ductility measures generally consist of the rotation, deflection, and curvature. Here, μ denotes the ductility index of the member, ∅ represents the rotation coefficient of the member, θ signifies the curvature of the member, and Δ denotes the deflection of the member.

### 6.2. Ductility Evaluation Using Displacement

Table 6 shows the ductility index values of the specimens. In this study, the ductility index of the structure was determined based on the displacements of the yield load and ultimate load. The ductility index values of the non-strengthened specimens R4C and R2C were 4.39 and 7.32, respectively, and their average ductility index was 5.9. Furthermore, the average ductility-index values obtained when using the NSM(P), NSM(N), EP, EB, and SE methods equaled 4.73, 3, 1.7, 1.87, and 4, respectively. Thus, all methods yield a ductility index that is lower compared to that of the non-strengthened specimens. However, the NSM and SE methods yield ductility indices that exceed those obtained using the EP and EB methods by up to two times or more.

### 6.3. Ductility of the Energy Ratio Method

The second method to derive the ductility index is to use the energy concept that was proposed by Jeong. In general, RC and steel structures have clear definitions of the yield state [28]. However, FRP structures do not have a clearly defined yield point. Thus, as shown in Figure 13, the ductility is derived by applying the ductility index Equations (2)–(4) by introducing the energy concept that was proposed by Jeong [33]:(2)$$E_{tot}=E_{inel}+E_{el}$$
(3)S=(P1S1+(P2−P1)S2+(P3−P2)S3)/P3
(4)$$\mu =\left(\frac{E_{tot}}{E_{el}}+1\right)/2$$

Table 7 shows the ductility values that are based on the energy ratio. The ductility values of the non-strengthened specimens R4C and R2C are 93.47 and 93.51, respectively, and the average ductility is 93.49. Furthermore, the average ductility values of NSM(P), NSM(N), EP, EB, and SE are 87.15, 67.62, 73.63, 63.08, and 98.83, respectively. Thus, every method except SE shows a ductility value that is lower compared to that obtained for the non-strengthened specimen. In addition, the overall energy ductility shows a similar trend as the ductility index that is based on deflection. Finally, according to the ductility grade that was suggested by Grace, the ductile, semi-ductile, and brittle sections can be classified as energy ratios of 75% or higher, 70–74%, and 69% or lower, respectively [35].

## 7. Conclusions

This study analyzed the design values and flexural test results by applying a variety of strengthening materials, including CFRP, GFRP, and a strand to the strengthening methods of NSM, EP, EB, and SE. In addition, the structural performance and stiffness were comparatively analyzed. Furthermore, the ductility index variations of the general and strengthened RC structures were examined based on the experimental values. Additionally, some problems of these construction methods can be observed in this study. As a result, the following conclusions were derived:(1)The NSM method showed highly similar design and experimental values of the maximum load, whereas the EP, EB, and SE methods showed differences of –60–40% between the design and experimental values. The reason for these differences is that the reinforcement design is not clearly defined in South Korea and the structures did not exhibit their performance properly. This is because the material and structure could not behave in an integral manner due to a construction problem when applying the various methods.(2)Regarding the stiffness increasing effect, every method showed an improving trend. In particular, in cases involving high concrete strength and action of a prestress force, such as those involving use of the EP and NSM(P) methods, the stiffness almost doubled compared to the non-strengthened specimen. However, the EP and EB methods failed the structure immediately after yielding, whereas the NSM method showed sufficient ductility after yielding.(3)In the comparison of the load–displacement curve, under the condition of concrete that was subjected to 40 MPa, the maximum load and displacement of the R4NSP specimen were 337.7 kN and 155.98 mm, respectively, which were approximately 106% and 90% higher, respectively, than those of the non-strengthened specimens. In cases involving 20-MPa concrete strength, the R2EBN and R2EPP specimens could sustain greater loads compared to the non-strengthened specimens, but they showed highly brittle behaviors, which indicates that the R2NSP specimen was appropriate. Therefore, by considering the stiffness, maximum load, and maximum displacement in this study, the most efficient strengthening method for the RC structures is NSM.(4)The strain of the steel bar was 0.002, whereas the compressive failure strain of concrete for the prevention of brittle failure was 0.003. Thus, with respect to the strain of the steel bar, the R4EEP specimen demonstrated the highest yield strength of 270 kN, while R2NSN(S) exhibited the lowest yield strength of 154.2 kN. Furthermore, in the case of concrete, every specimen reached the maximum load before reaching the compressive failure strain of 0.003.(5)With respect to the ductility evaluation methods, the ductility index based on the deflection and the ductility based on the energy ratio were compared. As a result, the NSM and SE methods had approximately twice or more ductility indices than the EP and EB methods. In addition, NSM and SE also showed excellent ductility. This result confirms that the ductility index based on the deflection and the ductility based on the energy ratio have similar trends.(6)The most critical advantages of the NSM, EP, EB, and SE methods determined from this study are as follows: the NSM method protects the reinforcement material with the burial effect, the EP method introduces external force, the EB method enables easy construction, and the SE method increases the bending moment due to the rise of the neutral axis. Improvements in actual structural performance by applying these methods can be confirmed.(7)Accurate designs for the various strengthening methods have not yet been clearly defined. Common problems in construction includes anchorages, the interface between the member and the material, the step difference between the member and anchorage, the integration behavior with the member, and the material properties differ from manufacturer to manufacturer. Therefore, to solve these problems in future studies, more accurate data should be secured and reflected in the design and the actual construction sites. This can be achieved by setting higher strengthening values, considering more strengthening methods, and using a greater number of material parameters from the strengthening companies.(8)In this study, the experiment was limited to only two types of FRP: CFRP and GFRP. However, in future studies, aramid fibers will be further considered, and comparative analysis will be performed on the same structural method and material-specific experiments. Since all fibers have unique material properties and many factors directly affect the structure such as elastic modulus, more precise research will be required. Additionally, based on the information, it will be possible to reinforce structure economically, safely, and sustainably.

## Figures and Tables

**Figure 1 polymers-13-00780-f001:**
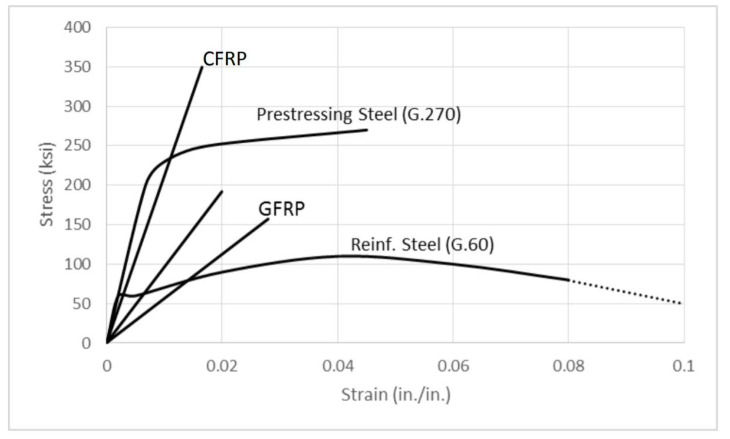
Typical stress–strain curves for steel and FRP (Fiber-Reinforced Polymer) [32].

**Figure 2 polymers-13-00780-f002:**
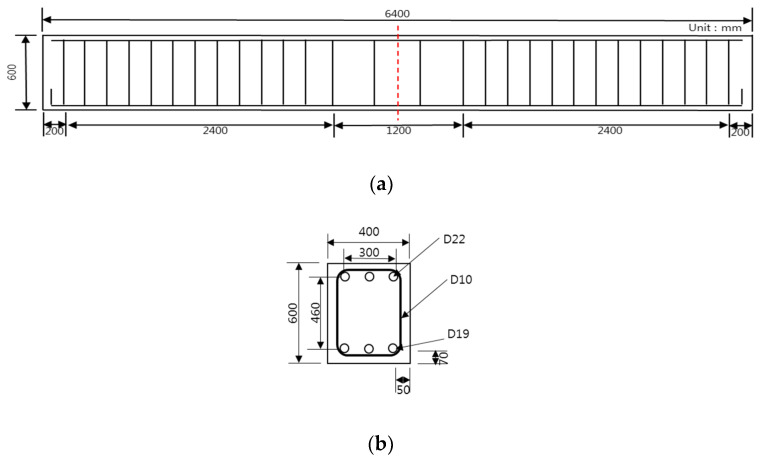
Details of the RC beam: (**a**) RC specimen drawing; (**b**) cross-sectional details of RC (D19) specimen.

**Figure 3 polymers-13-00780-f003:**
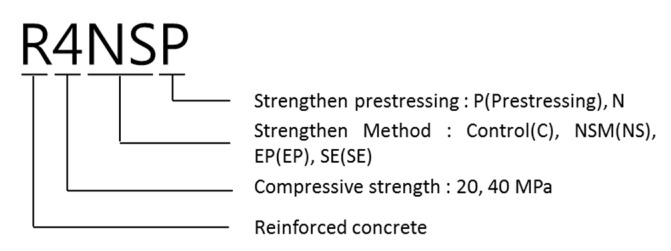
Specimen nomenclature.

**Figure 4 polymers-13-00780-f004:**
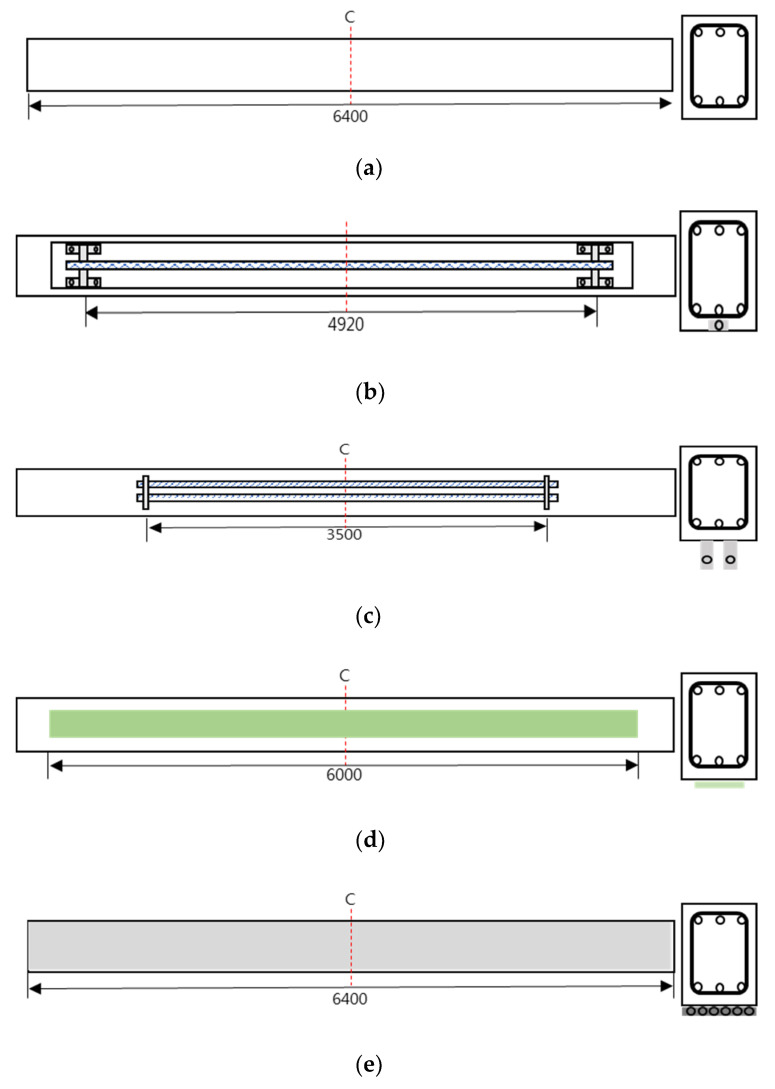
Strengthening methods: (**a**) control; (**b**) near-surface mounted (NSM); (**c**) external prestressing (EP); (**d**) external bonding (EB); and (**e**) section enlargement (SE).

**Figure 5 polymers-13-00780-f005:**
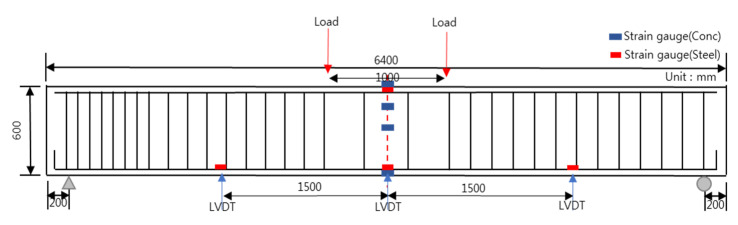
Flexure experiment method.

**Figure 6 polymers-13-00780-f006:**
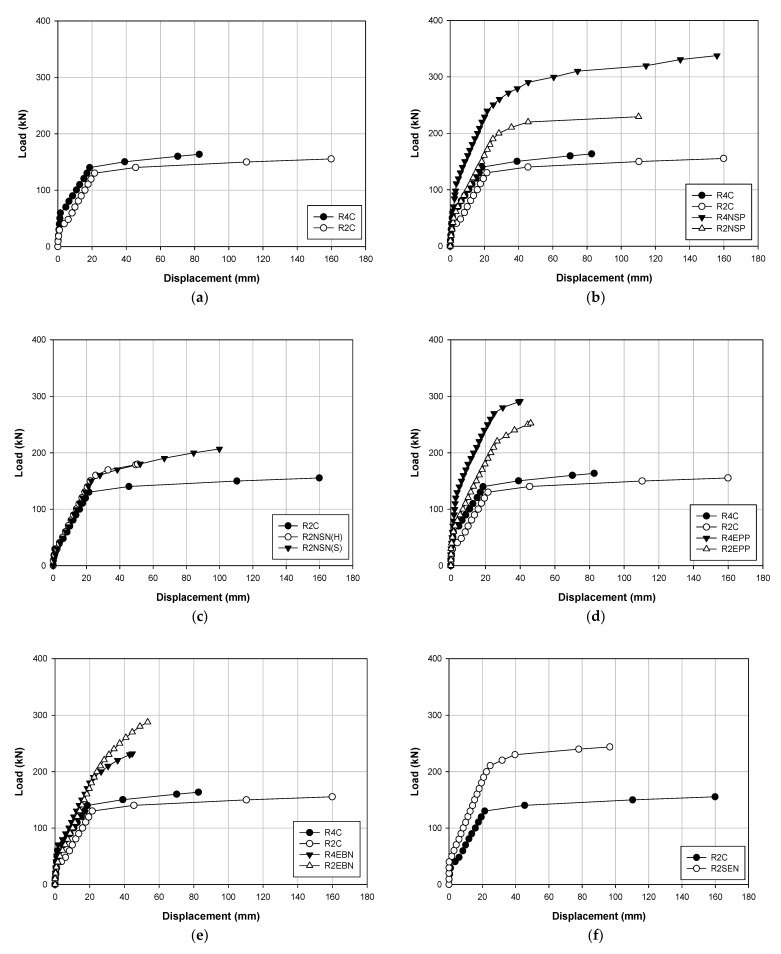
Load–displacement curves corresponding to (**a**) the RC control; (**b**) NSM(P); (**c**) NSM(N); (**d**) EP(P); (**e**) EB; and (**f**) SE strengthening methods.

**Figure 7 polymers-13-00780-f007:**
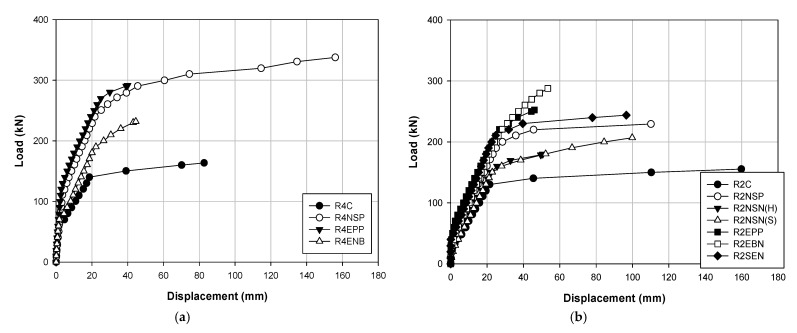
Load–displacement curves corresponding to different strengthening methods obtained for concrete strengths: (**a**) 40 MPa; (**b**) 20 MPa.

**Figure 8 polymers-13-00780-f008:**
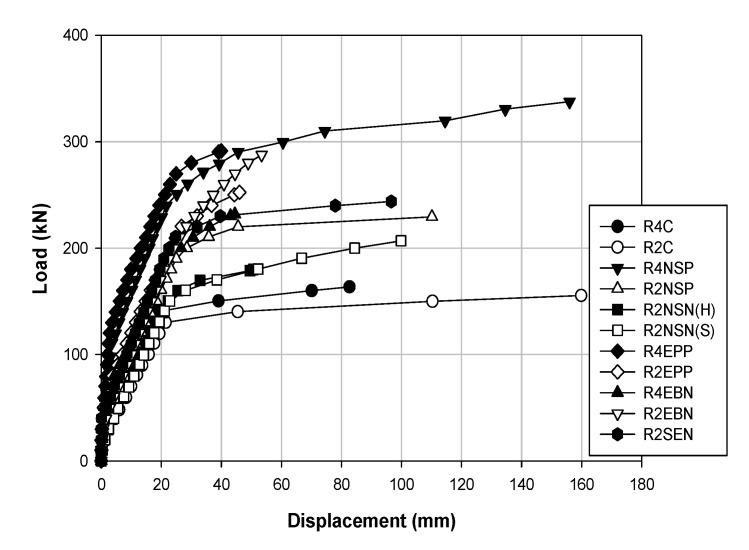
Load–displacement curves for all specimens.

**Figure 9 polymers-13-00780-f009:**
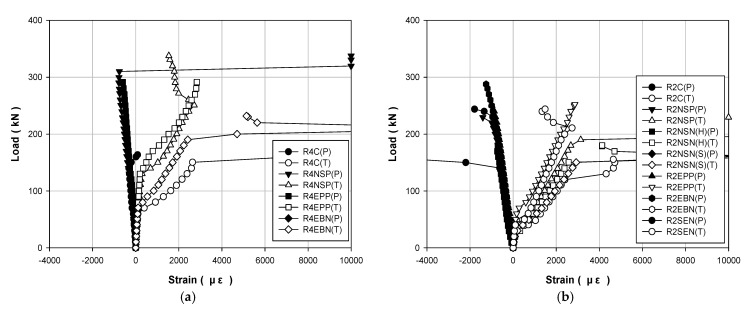
Load–strain curves for steel-bar specimens strengthened using different methods under different concrete strengths: (**a**) 40 MPa; (**b**) 20 MPa.

**Figure 10 polymers-13-00780-f010:**
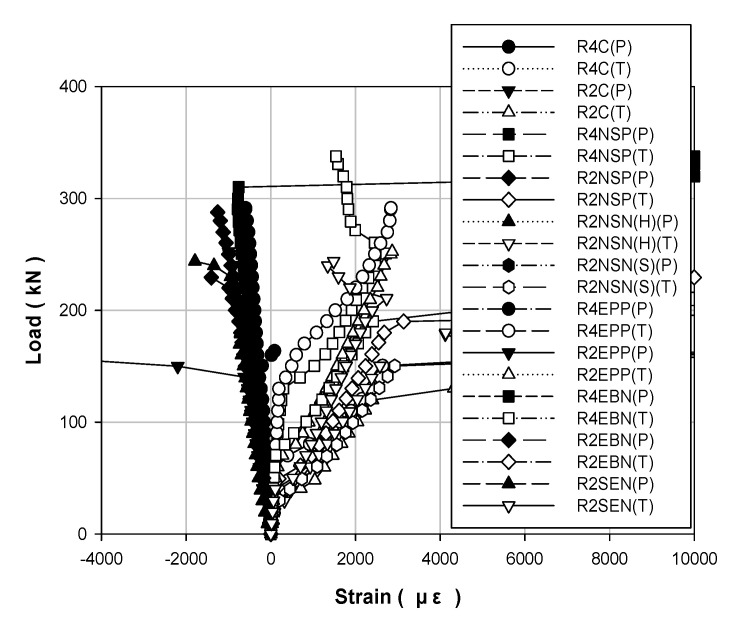
Load–strain curves for all steel-bar specimens.

**Figure 11 polymers-13-00780-f011:**
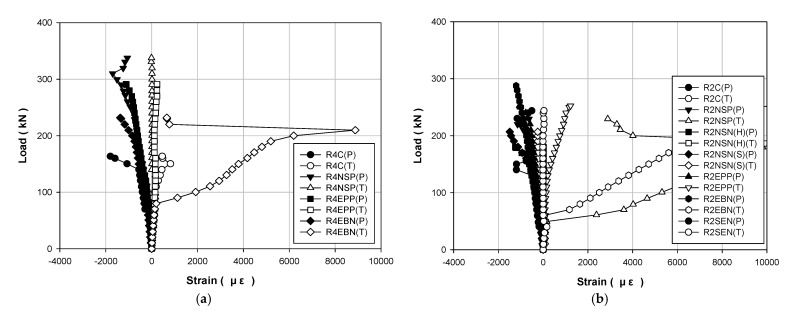
Load–strain curves for concrete specimens strengthened using different methods under different concrete strengths: (**a**) 40 MPa; (**b**) 20 MPa.

**Figure 12 polymers-13-00780-f012:**
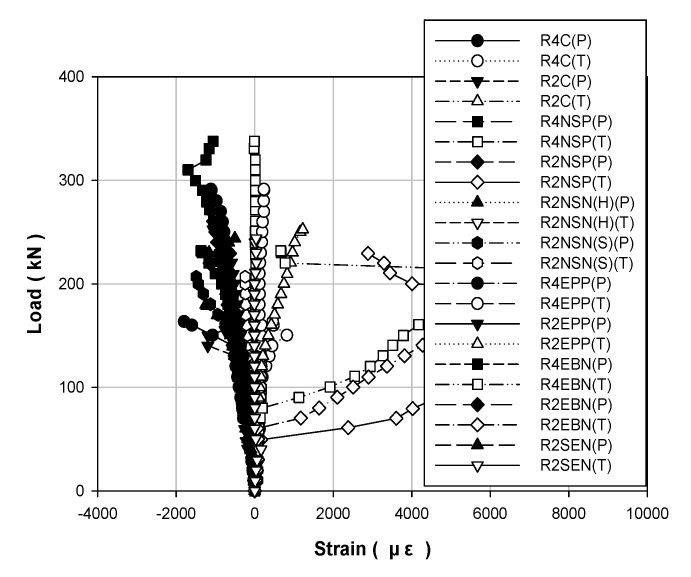
Load–strain curves for all concrete specimens.

**Figure 13 polymers-13-00780-f013:**
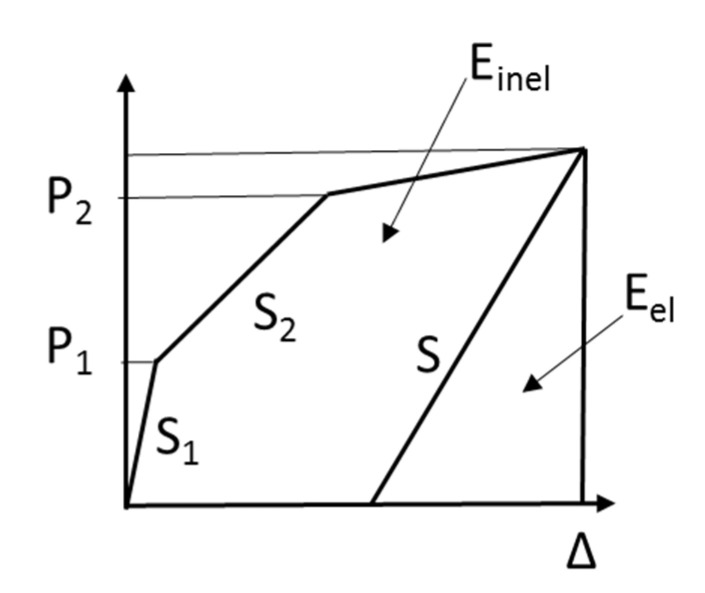
Ductility of energy ratio.

**Table 1 polymers-13-00780-t001:** Concrete mixture proportion.

Unit Weight (kg/m^3^)							Compressive Strength (MPa)
CE	W	C	F	FA	GGBS	AE	
265	162	905	954	30	-	2.3	20
258	151	1039	626	77	180	5.3	40

Abbreviations: CE, cement; W, water; C, coarse aggregate; F, fine aggregate; FA, fly ash; GGBS, ground granulated blast-furnace slag; AE, air-entraining agent.

**Table 2 polymers-13-00780-t002:** Test specimen details.

Specimens	Strengthening Methods	Strengthening Material	Strengthening Amount (EA)
R4C	-	-	-
R2C	-	-	-
R4NSP	Near Surface Mounted	CFRP	1
R2NSP	Near Surface Mounted	CFRP	1
R2NSN(H)	Near Surface Mounted	CFRP	1
R4NSN(S)	Near Surface Mounted	CFRP	1
R4EPP	External Prestressing	Strand	2
R2EPP	External Prestressing	Strand	2
R4EBN	External Bonding	CFRP Sheet	2
R2EBN	External Bonding	CFRP Sheet	2
R2SEN	Section Enlargement	GFRP bar	6

**Table 3 polymers-13-00780-t003:** Properties of the strengthening materials.

Material Property	Steel Bar	CFRP Bar	PS Strands
Young’s Modulus (GPa)	200	165	200
Yield Stress (MPa)	400	-	1597.9
Ultimate Stress (MPa)	560	2750	1880.7

**Table 4 polymers-13-00780-t004:** Comparison between analytical and experimental values of crack, yield, and ultimate loads.

Specimens	$\mathrm{Crack}\;\mathrm{Load}\;(P_{cr})$ (kN)		$\mathrm{Yield}\;\mathrm{Load}\;(P_{y})$ (kN)		$\mathrm{Ultimate}\;\mathrm{Load}\;(P_{u})$ (kN)	
	Analytical	Experimental	Analytical	Experimental	Analytical	Experimental
R4C	55.8	57.7	116.7	143.1	128.8	163.5
R2C	33.4	31.2	115.7	134.4	118.9	155.4
R4NSP	119.8	110.0	231.6	256.4	338.0	337.7
R2NSP	65.4	57.1	172.3	200.1	222.0	229.5
R2NSN(H)	33.4	23.6	121.7	158.8	192.0	180.1
R2NSN(S)	33.4	22.8	121.3	154.2	193.0	206.9
R4EPP	117.2	128.9	247.5	270.0	205.7	291.4
R2EPP	57.8	58.9	199.2	214.3	200.2	252.5
R4EBN	55.8	69.5	158.3	194.6	695.5	232.0
R2EBN	33.4	37.2	156.8	216.4	516.9	287.7
R2SEN	33.4	39.1	170.1	210.1	376.3	243.9

**Table 5 polymers-13-00780-t005:** Load–displacement results.

Specimens	$\mathrm{Crack}\;(P_{cr})$		$\mathrm{Yield}\;(P_{y})$		$\mathrm{Ultimate}\;(P_{u})$	
	Load (kN)	Displacement (mm)	Load (kN)	Displacement (mm)	Load (kN)	Displacement (mm)
R4C	57.7	1.35	143.1	18.85	163.5	82.68
R2C	31.2	1.2	134.4	21.84	155.4	159.87
R4NSP	110.0	3.55	256.4	27.97	337.7	155.98
R2NSP	57.1	3.04	200.1	28.44	229.5	110.16
R2NSN(H)	23.6	1.42	158.8	25.01	180.1	50.61
R2NSN(S)	22.8	1.87	154.2	25.11	206.9	99.84
R4EPP	128.9	3.54	270.0	24.94	291.4	39.93
R2EPP	58.9	1.37	214.3	25.72	252.5	46.05
R4EBN	69.5	1.70	194.6	24.41	232.0	44.49
R2EBN	37.2	1.38	216.4	27.87	287.7	53.40
R2SEN	39.1	0.05	210.1	24.15	243.9	96.63

**Table 6 polymers-13-00780-t006:** Ductility evaluation of deflection.

Specimens	$\mathrm{Yield}\;(P_{y})$		$\mathrm{Ultimate}\;(P_{u})$		Ductility Index
	Load (kN)	Displacement (mm)	Load kN)	Displacement (mm)	
R4C	143.1	18.85	163.5	82.68	4.39
R2C	134.4	21.84	155.4	159.87	7.32
R4NSP	256.4	27.97	337.7	155.98	5.58
R2NSP	200.1	28.44	229.5	110.16	3.87
R2NSN(H)	158.8	25.01	180.1	50.61	2.02
R2NSN(S)	154.2	25.11	206.9	99.84	3.98
R4EPP	270.0	24.94	291.4	39.93	1.60
R2EPP	214.3	25.72	252.5	46.05	1.79
R4EBN	194.6	24.41	232.0	44.49	1.82
R2EBN	216.4	27.87	287.7	53.40	1.92
R2SEN	210.1	24.15	243.9	96.63	4.00

**Table 7 polymers-13-00780-t007:** Results obtained using ductility-of-energy-ratio method.

Specimens	Ductility Index	Energy				Analysis
		Total	Elastic	Inelastic	Rate	
R4C	4.39	11,581.09	756.33	10,824.75	93.47	Ductile
R2C	7.32	21,728.26	1410.40	20,317.86	93.51	Ductile
R4NSP	5.58	42,694.36	4439.12	38,255.24	89.60	Ductile
R2NSP	3.87	20,906.69	3200.95	17,705.73	84.69	Ductile
R2NSN(H)	2.02	6506.08	2465.26	4040.83	62.11	Brittle
R4NSN(S)	3.98	15,570.56	4185.30	11,385.26	73.12	Semi-Ductile
R4EPP	1.60	8704.08	2188.00	6516.07	74.86	Semi-Ductile
R2EPP	1.79	8111.58	2238.51	5873.07	72.40	Semi-Ductile
R4EBN	1.82	7340.99	1734.28	5606.72	76.38	Ductile
R4EBN	1.92	9819.44	4932.00	4887.44	49.77	Brittle
R4SEN	4.00	19,456.80	228.09	19,228.71	98.83	Ductile

## Data Availability

The data presented in this study are available on request from the corresponding author.

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
