# Peer review of "Structural Behavior Evaluation of Reinforced Concrete Using the Fiber-Reinforced Polymer Strengthening Method"

_polymers, 2021, doi:10.3390/polym13050780_

Round 1

Reviewer 1 Report

Dear Authors,

I have read the manuscript with interest and some questions raised. Enlisted please find my comments.

Overall. General English grammar revision (Minor spelling errors).

Key words. “structural polymer” and “nanofiller” could be added in my opinion.

Introduction. Authors stated “However, concrete is a brittle material and its disadvantages include cracking, partial breakage, and its heavy weight.”. Please add a reference for this statement.

Introduction. Authors stated “In addition, to realize material strengthening, fiber-reinforced polymers (FRPs) can replace steels, the use of which is reported in existing literature.”. Please point out better the role of fiber reinforcement in existing literature. Authors could add that " FRP are axial particulates embedded in fitting matrices. The advantages of FRCs over unreinforced materials are well known and their characteristics are useful for many scopes. The fields with a constantly increasing number of studies about FRC applications are Aerospace (Fibre reinforced composites in aircraft construction. Soutis C. Progress in Aerospace Sciences Volume 41, Issue 2, February 2005, Pages 143-151), medicine (Fiber-Reinforced Composites for Dental Applications. Scribante A, Vallittu PK, Özcan M. Biomed Res Int. 2018 Nov 1;2018:4734986.) and construction Science (Thermal Stability, Fire Performance, and Mechanical Properties of Natural Fibre Fabric-Reinforced Polymer Composites with Different Fire Retardants; Erik Valentine Bachtiar, Katarzyna Kurkowiak , Libo Yan , Bohumil Kasal and Torsten Kolb. Polymers 2019, 11(4), 699).” These considerations could be added in Introduction Section.

Figure 1. Please enlarge to increase readability.

Figure 1. This figure reports Typical stress–strain curves for steel and FRP. Please add if the graph is from personal data of the Authors or if it derives from a published textbook or manuscript (in this latter case, please add reference and verify copyright permissions)

Background of FRP. Authors stated “In the history of composite materials, FRPs were developed over six decades ago for applications in leisure activities. With the development of material technology and the passage of time, fibers such as aramid, carbon, and glass have been developed for high functionality, mechanical properties, high durability, and weight reduction.”. Please add a reference for this statement.

Background of FRP. Authors stated “However, from the late 1980s to early 1990s, the defense market suffered a recession, but the prices of composite materials were further reduced by the virtue of continued growth. Consequently, developed countries around the world expanded the applications of composite materials to infrastructures from the mid to late 1990s.”. Please add a reference for this statement.

Introduction. Authors stated “Composite materials refer to the combination of two or more types of materials. Conceptually, composite materials are artificially manufactured materials with a higher performance than the existing materials while physically and chemically maintaining the original phase of each material, even after being combined with two or more types of 110 materials.”. Please add a reference for this statement.

Background of FRP. Authors stated “Although many studies have been conducted on FRPs, FRPs”. Please add at least a couple (or more) of these studies as reference.

Background of FRP. Authors stated “CFRP is a material that exhibits high elasticity and strength by adding carbon fibers to plastics, and carbon fiber composite materials are made of carbon fiber as a reinforcing material and a matrix resin that is combined with plastics.”. Please add a reference for this statement.

Background of FRP. Authors stated “In comparison to iron, CFRP has a strength that is 10 times higher and an elastic modulus that is seven times higher, but its weight is only 25% of iron”. Please add a reference for this statement.

Background of FRP. Authors stated “GFRP is a plastic that is reinforced with glass fiber and it is processed by an unsaturated polyester with a diameter of 0.1 mm or less. GFRP is lighter than aluminum, and it is stronger, lighter, and harder than general iron; thus, it can resist an external impact and it has a very high tensile strength.”. Please add a reference for this statement.

Reinforcement Techniques. Authors stated “NSM is made by digging grooves and placing reinforcements in the existing structures. In the past, reinforcement was mainly carried out with steel bars, but reinforcements that use FRP materials have been recently applied.”. Please add a reference for this statement.

Reinforcement Techniques. Authors stated “EP is a method that introduces a prestress force by installing anchorages on existing structures and it places new PS tension members to improve the stress state and load carrying capacity of the existing structures”. Please add a reference for this statement.

Reinforcement Techniques. Authors stated “It has the advantage of allowing smooth transportation in the process of bridge reinforcement.”. Please add a reference for this statement.

Reinforcement Techniques. Authors stated “SE is applied to concrete bridges for increasing the bending moment that accompanies the rise of the neutral axis by improving the load-bearing performance against compression and shearing of the top plate.”. Please add a reference for this statement.

Reinforcement Techniques. Authors stated “Table 1 lists the concrete mix proportions. The compressive strengths of concrete after 28 days are 20 and 40 MPa”. Please add a reference for this statement.

Table 1. Is this table derived from literature or from Authors’ data? Please point out.

Test Conditions and Methods. Authors stated “There are 11 specimens in total, which are composed of two control specimens and nine reinforced specimens”. Please add if and how sample size calculation has been performed.

Test Conditions and Methods. Authors stated “A TDS-530 data logger was used for the measurements.”. Please add details about data logger used: manufacturer, City and State.

Test Conditions and Methods. Authors stated “Regarding the loading method, the load was controlled to 30 mm at a rate of 0.03 mm/s, and a displacement control of 0.1 mm/sec was performed later”. Please add a reference for this method.

Test Conditions and Methods. Authors stated “The universal testing machine (UTM) that was used in this study was a 200 ton-grade device”. Please add details about data logger used: manufacturer, City and State.

Test Conditions and Methods. Please add the software used for graphical representation of the results, along with manufacturer, City and State. Additionally, was any statistical test applied? In the case, please point out software used.

Conclusions. Please add a sentence showing the limitations of the present report.

Conclusions. Provide a general interpretation of the results in the context of other evidence, and implications for future research.

References. Some references are quite old (1996; 1994; 1998). If possible, please switch with some more modern research.

Author Response

Response to Reviewer 1 Comments

Thank you for giving me the opportunity to submit a revised draft of my manuscript titled “Structural-behavior Evaluation of Reinforced Concrete using Fiber-reinforced Polymer Strengthening Method” to Polymers. We appreciate the time and effort that you and the reviewers have dedicated to providing your valuable feedback on our manuscript. We are grateful to the reviewers for their insightful comments on our paper. We have been able to incorporate changes to reflect most of the suggestions provided by the reviewers.

Also, language editing for the manuscript has been done. We have used the services of a professional editing company for the same.

Comment 1: General English grammar revision (Minor spelling errors).

Response: We have revised the text to reflect reviewers’ comment.

Comment 2: Key words. “structural polymer” and “nanofiller” could be added in my opinion.

Response: We have revised the text to reflect reviewers’ comment. (line 24)

Comment 3: Introduction. Authors stated “However, concrete is a brittle material and its disadvantages include cracking, partial breakage, and its heavy weight.”. Please add a reference for this statement.

Response: We have revised the text to reflect reviewers’ comment. (line 30)

Comment 4: Introduction. Authors stated “In addition, to realize material strengthening, fiber-reinforced polymers (FRPs) can replace steels, the use of which is reported in existing literature.”. Please point out better the role of fiber reinforcement in existing literature. Authors could add that "FRP are axial particulates embedded in fitting matrices. The advantages of FRCs over unreinforced materials are well known and their characteristics are useful for many scopes. The fields with a constantly increasing number of studies about FRC applications are Aerospace (Fibre reinforced composites in aircraft construction. Soutis C. Progress in Aerospace Sciences Volume 41, Issue 2, February 2005, Pages 143-151), medicine (Fiber-Reinforced Composites for Dental Applications. Scribante A, Vallittu PK, Özcan M. Biomed Res Int. 2018 Nov 1;2018:4734986.) and construction Science (Thermal Stability, Fire Performance, and Mechanical Properties of Natural Fibre Fabric-Reinforced Polymer Composites with Different Fire Retardants; Erik Valentine Bachtiar, Katarzyna Kurkowiak , Libo Yan, Bohumil Kasal and Torsten Kolb. Polymers 2019, 11(4), 699).” These considerations could be added in Introduction Section.

Response: We have revised the text to reflect reviewers’ comment. (line 67-70)

Comment 5: Figure 1. Please enlarge to increase readability.

Response: We have revised the text to reflect reviewers’ comment. (line 101)

Comment 6: Figure 1. This figure reports Typical stress–strain curves for steel and FRP. Please add if the graph is from personal data of the Authors or if it derives from a published textbook or manuscript (in this latter case, please add reference and verify copyright permissions)

Response: We have revised the text to reflect reviewers’ comment. (line 102)

Comment 7: Background of FRP. Authors stated “In the history of composite materials, FRPs were developed over six decades ago for applications in leisure activities. With the development of material technology and the passage of time, fibers such as aramid, carbon, and glass have been developed for high functionality, mechanical properties, high durability, and weight reduction.”. Please add a reference for this statement.

Response: We have revised the text to reflect reviewers’ comment. (line 107)

Comment 8: Background of FRP. Authors stated “However, from the late 1980s to early 1990s, the defense market suffered a recession, but the prices of composite materials were further reduced by the virtue of continued growth. Consequently, developed countries around the world expanded the applications of composite materials to infrastructures from the mid to late 1990s.”. Please add a reference for this statement.

Response: We have revised the text to reflect reviewers’ comment. (line 115)

Comment 9: Background of FRP. Authors stated “Composite materials refer to the combination of two or more types of materials. Conceptually, composite materials are artificially manufactured materials with a higher performance than the existing materials while physically and chemically maintaining the original phase of each material, even after being combined with two or more types of materials.”. Please add a reference for this statement.

Response: We have revised the text to reflect reviewers’ comment. (line 124)

Comment 10: Background of FRP. Authors stated “Although many studies have been conducted on FRPs, FRPs”. Please add at least a couple (or more) of these studies as reference.

Response: We have revised the text to reflect reviewers’ comment. (line 133)

Comment 11: Background of FRP. Authors stated “CFRP is a material that exhibits high elasticity and strength by adding carbon fibers to plastics, and carbon fiber composite materials are made of carbon fiber as a reinforcing material and a matrix resin that is combined with plastics.”. Please add a reference for this statement.

Response: We have revised the text to reflect reviewers’ comment. (line 137)

Comment 12: Background of FRP. Authors stated “In comparison to iron, CFRP has a strength that is 10 times higher and an elastic modulus that is seven times higher, but its weight is only 25% of iron”. Please add a reference for this statement.

Response: We have revised the text to reflect reviewers’ comment. (line 139)

Comment 13: Background of FRP. Authors stated “GFRP is a plastic that is reinforced with glass fiber and it is processed by an unsaturated polyester with a diameter of 0.1 mm or less. GFRP is lighter than aluminum, and it is stronger, lighter, and harder than general iron; thus, it can resist an external impact and it has a very high tensile strength.”. Please add a reference for this statement.

Response: We have revised the text to reflect reviewers’ comment. (line 150)

Comment 14: Reinforcement Techniques. Authors stated “NSM is made by digging grooves and placing reinforcements in the existing structures. In the past, reinforcement was mainly carried out with steel bars, but reinforcements that use FRP materials have been recently applied.”. Please add a reference for this statement.

Response: We have revised the text to reflect reviewers’ comment. (line 158)

Comment 15: Reinforcement Techniques. Authors stated “EP is a method that introduces a prestress force by installing anchorages on existing structures and it places new PS tension members to improve the stress state and load carrying capacity of the existing structures”. Please add a reference for this statement.

Response: We have revised the text to reflect reviewers’ comment. (line 168)

Comment 16: Reinforcement Techniques. Authors stated “It has the advantage of allowing smooth transportation in the process of bridge reinforcement.”. Please add a reference for this statement.

Response: We have revised the text to reflect reviewers’ comment. (line 174)

Comment 17: Reinforcement Techniques. Authors stated “SE is applied to concrete bridges for increasing the bending moment that accompanies the rise of the neutral axis by improving the load-bearing performance against compression and shearing of the top plate.”. Please add a reference for this statement.

Response: We have revised the text to reflect reviewers’ comment. (line 184)

Comment 18: Reinforcement Techniques. Authors stated “Table 1 lists the concrete mix proportions. The compressive strengths of concrete after 28 days are 20 and 40 MPa”. Please add a reference for this statement. (Table 1. Is this table derived from literature or from Authors’ data? Please point out.)

Response: We have revised the text to reflect reviewers’ comment. (line 192)

Comment 19: Test Conditions and Methods. Authors stated “A TDS-530 data logger was used for the measurements.”. Please add details about data logger used: manufacturer, City and State.

Response: We have revised the text to reflect reviewers’ comment. (line 247)

Comment 20: Test Conditions and Methods. Authors stated “Regarding the loading method, the load was controlled to 30 mm at a rate of 0.03 mm/s, and a displacement control of 0.1 mm/sec was performed later”. Please add a reference for this method.

Response: We have revised the text to reflect reviewers’ comment. (line 256)- In addition, the load was controlled to 30 mm at a rate of 0.03 mm/s, and a displacement control of 0.1 mm/sec was performed later.

Comment 21: Test Conditions and Methods. Authors stated “The universal testing machine (UTM) that was used in this study was a 200 ton-grade device”. Please add details about data logger used: manufacturer, City and State.

Response: We have revised the text to reflect reviewers’ comment. (line 258)

Comment 22: Test Conditions and Methods. Please add the software used for graphical representation of the results, along with manufacturer, City and State. Additionally, was any statistical test applied? In the case, please point out software used.

Response: We have revised the text to reflect reviewers’ comment. (line 259)

Comment 23: Conclusions. Please add a sentence showing the limitations of the present report.

Response: We have revised the text to reflect reviewers’ comment. (line 495)

Comment 24: Conclusions. Provide a general interpretation of the results in the context of other evidence, and implications for future research.

Response: We have revised the text to reflect reviewers’ comment. (line 547)

Comment 25: References. Some references are quite old (1996; 1994; 1998). If possible, please switch with some more modern research.

Response: We have revised the text to reflect reviewers’ comment. (line 603, 631, 634)

Reviewer 2 Report

Reinforced concrete (RC) structures are being applied worldwide to a variety of infrastructures owing to their excellent structural performance, durability, fire resistance, and economic efficiency. The cracking, stress, and strain of reinforced concrete appears to be different depending on the member type, applied load, member shape, dimensions, the arrangement of the steel bars, and the properties of the concrete and steel bars. There are various reinforcement methods for RC structures. In this paper, this investigation selected four reinforcement methods: near-surface mounting (NSM), external prestressing (EP), external bonding (EB), and section enlargement (SE). Various strengthening materials such as a CFRP, GFRP, and prestressing (PS) strand were applied to the NSM, EP, EB, and SE methods, followed by flexural experiments. In addition, changes in the ductility of the RC structures were examined. The concrete EP and NSM(P) methods have a stiffness that is almost double than the non-strengthened specimen. I am pleased to send you moderate comments. The results and theme of this paper is quite interesting. The layout is clear and easy to understand. Generally, this manuscript makes fair impression and my recommendation is that it merits publication in this Journal, after the following major revision:

  1. The authors need to reorganize the current introduction, which normally consists of three parts at least: background, literature review, brief of the proposed work. The current one is nothing but a literature review. Why their work is important comparing to previous reports? I think this is essential to keep the interest of the reader.
  2. The concrete EP and NSM(P) methods have a stiffness that is almost double than the non-strengthened specimen. However, because the EP and EB methods are brittle, the NSM(P) method with ductile behavior is considered the most effective. The authors should give some explanation on above results and analyze the physical mechanism in detail.
  3. In Fig.6-12, the authors should give the explanations for the difference of data collected from different sources.
  4. Experiment part. Although the results look “making sense”, the current form reads like a simple lab report. The authors should dig deeper in the results by presenting some in-depth discussion.
  5. In order to verify the validity of current results, authors need to compare their results with reported experimental data and models in literatures.
  6. It is suggested to discuss what the main advantages the proposed NSM, EP, EB, and SE methods has.
  7. Fiber reinforced polymer composite including fibrous porous media has been widely used in many fields of life. Besides NSM, EP, EB, and SE methods, fractal method is a very important tool, which can investigate the physical properties of fiber reinforced polymer composite including fibrous porous media (see [A fractal model for capillary flow through a single tortuous capillary with roughened surfaces in fibrous porous media, Fractals, 2021, 28(4):2150017; International Journal of Heat and Mass Transfer, 2019, 137:365-371]). Authors should introduce some related knowledge to readers.
  8. Please, expand the conclusions in relation to the specific goals and the future work.

Author Response

Response to Reviewer 2 Comments

Thank you for giving me the opportunity to submit a revised draft of my manuscript titled “Structural-behavior Evaluation of Reinforced Concrete using Fiber-reinforced Polymer Strengthening Method” to Polymers. We appreciate the time and effort that you and the reviewers have dedicated to providing your valuable feedback on our manuscript. We are grateful to the reviewers for their insightful comments on our paper. We have been able to incorporate changes to reflect most of the suggestions provided by the reviewers.

Also, language editing for the manuscript has been done. We have used the services of a professional editing company for the same.

Comment 1: The authors need to reorganize the current introduction, which normally consists of three parts at least: background, literature review, brief of the proposed work. The current one is nothing but a literature review. Why their work is important comparing to previous reports? I think this is essential to keep the interest of the reader.

Response: We have revised the text to reflect reviewers’ comment. (background : line 27-51, literature review : line 52-89, brief of the proposed work : line 90-100)

Comment 2: The concrete EP and NSM(P) methods have a stiffness that is almost double than the non-strengthened specimen. However, because the EP and EB methods are brittle, the NSM(P) method with ductile behavior is considered the most effective. The authors should give some explanation on above results and analyze the physical mechanism in detail.

Response: We have revised the text to reflect reviewers’ comment. (line 373-387)

Comment 3: In Fig.6-12, the authors should give the explanations for the difference of data collected from different sources.

Response: We have revised the text to reflect reviewers’ comment. (line 264-269)

Comment 4: Experiment part. Although the results look “making sense”, the current form reads like a simple lab report. The authors should dig deeper in the results by presenting some in-depth discussion.

Response: We have revised the text to reflect reviewers’ comment. (line 424-444)

Comment 5: In order to verify the validity of current results, authors need to compare their results with reported experimental data and models in literatures.

Response: We have revised the text to reflect reviewers’ comment. (line 340-351)

Comment 6: It is suggested to discuss what the main advantages the proposed NSM, EP, EB, and SE methods has.

Response: We have revised the text to reflect reviewers’ comment. (line 95-100, 532-537 )

Comment 7: Fiber reinforced polymer composite including fibrous porous media has been widely used in many fields of life. Besides NSM, EP, EB, and SE methods, fractal method is a very important tool, which can investigate the physical properties of fiber reinforced polymer composite including fibrous porous media (see [A fractal model for capillary flow through a single tortuous capillary with roughened surfaces in fibrous porous media, Fractals, 2021, 28(4):2150017; International Journal of Heat and Mass Transfer, 2019, 137:365-371]). Authors should introduce some related knowledge to readers.

Response: We have revised the text to reflect reviewers’ comment. (line 70-74, References 29, 30)

Comment 8: Please, expand the conclusions in relation to the specific goals and the future work.

Response: We have revised the text to reflect reviewers’ comment. (line 547)

Round 2

Reviewer 1 Report

Good job.

Reviewer 2 Report

In Ref. 29, “Fractals 2021, 28” should be corrected as “Fractals 2021, 29, 2150017”